# The Length of Hospital Stay of Patients with Venous Thromboembolism: A Cross-Sectional Study from Jordan

**DOI:** 10.3390/medicina59040727

**Published:** 2023-04-07

**Authors:** Haneen Amawi, Rasha M. Arabyat, Sayer Al-Azzam, Toqa AlZu’bi, Hamza Tayseer U’wais, Alaa M. Hammad, Ruba Amawi, Mohammad B. Nusair

**Affiliations:** 1Department of Pharmacy Practice and Clinical Pharmacy, Faculty of Pharmacy, Yarmouk University, Irbid 22110, Jordan; 2Department of Clinical Pharmacy, Faculty of Pharmacy, Jordan University of Science and Technology, Irbid 22110, Jordan; 3Department of Pharmacy, Faculty of Pharmacy, Al-Zaytoonah University of Jordan, Amman 11733, Jordan; 4The Ministry of Health, Amman 11118, Jordan; 5Department of Sociobehavioral and Administrative Pharmacy, College of Pharmacy, Nova Southeastern University, Fort Lauderdale, FL 33328, USA

**Keywords:** venous thromboembolism, length of stay, healthcare use, hospitalization, inpatient management

## Abstract

*Background and Objectives*: Venous thromboembolism is one of the leading causes of mortality and disability worldwide. Treatment with anticoagulation therapy is essential and requires a delicate approach to select the most appropriate option to improve patient outcomes, including the length of hospital stay (LOS). The aim of this study was to determine the LOS among patients with acute onset of VTE in several public hospitals in Jordan. *Materials and Methods:* In this study, we recruited hospitalized patients with a confirmed diagnosis of VTE. We reviewed the electronic medical records and charts of VTE admitted patients in addition to a detailed survey to collect the patients’ self-reported data. Hospital LOS was categorized into three levels: 1–3 days, 4–6 days, and ≥7 days. An ordered logistic regression model was used to study the significant predictors of LOS. *Results:* A total of 317 VTE patients were recruited, with 52.4% of them were male and 35.3% aged between 50 and 69 years. Most patients had a deep vein thrombosis (DVT) diagnosis (84.2%), and most of the VTE cases were admitted for the first-time (64.6%). The majority of the patients were smokers (57.2%), overweight/obese (66.3%), and hypertensive (59%). Most of the VTE patients received Warfarin overlapped with low molecular weight heparins as their treatment regimen (>70%). Almost half of the admitted VTE patients (45%) were hospitalized for at least 7 days. Longer LOS was significantly associated with hypertension. *Conclusions:* We recommend using therapies that have been proven to reduce hospital LOS, such as non-vitamin K antagonist oral anticoagulants or direct oral anticoagulants, to treat VTE patients in Jordan. Additionally, preventing and controlling comorbidities such as hypertension is essential.

## 1. Introduction

Venous thromboembolism (VTE) is a common cardiovascular disease that can manifest as deep vein thrombosis (DVT), pulmonary embolism (PE), or both [1,2]. It usually results in significant non-specific symptoms and health issues such as leg pain and swelling, shortness of breath, and chest pain [1,3,4]. VTE is one of the leading causes of mortality and disability worldwide [5]. It is considered as a public health concern due to its devastating long-term outcomes which include recurrence, post-thrombotic syndrome (PTS), major bleeding due to anticoagulant therapy, chronic thromboembolic pulmonary hypertension, and death. These outcomes place a significant burden on VTE patients and their families [1,3]. 

The incidence of VTE is increasing and is associated with age, immobility, recent hospitalization, cancer, genetic factors, lifestyle, and other comorbidities. Furthermore, the confirmation of VTE diagnosis is continuously increasing due to advanced imaging and diagnostic tools [2]. In the United States, the annual incidence rate is approximately 123 per 100,000 adults [2]. 

The treatment and management of VTE require a delicate multidisciplinary approach while following well-structured guidelines [5]. The main treatment arm for VTE is anticoagulation therapy. The mortality rate without using effective anticoagulants is high, with a 30-day mortality rate that could reach up to 30%. Anticoagulants include injectable agents such as Heparin, Low Molecular Weight Heparins (LMWH), and factor X antagonists. Anticoagulants also include oral agents such as the old vitamin K antagonist (Warfarin) which, for years, was the only approved oral anticoagulant available in clinical practice. However, several Direct Oral Anticoagulants (DOACs), which are also named as non-vitamin K antagonist oral anticoagulant(s) (NOAC(s)), are now available (Dabigatran, Rivaroxaban, Apixaban, and Edoxaban) with superior Pharmacokinetics (PK) and less variability when compared to Warfarin [6,7]. Thus, the DOACs are now preferred over Warfarin as first line agents to treat acute DVT and PE patients without cancer, according to the latest 2019 ESC Guidelines for the diagnosis and management of acute pulmonary embolism developed in collaboration with the European Respiratory Society (ERS) [8] and the 2016 CHEST guidelines [7]. 

The hospital length of stay (LOS) is a key clinical and performance outcome measure that is commonly used as an indicator of efficiency. It is defined as the average days that patients spend in the hospital from the time of admission until discharge [9,10]. Several studies confirmed that a significant portion of the days spent by patients in hospitals are inappropriate and unnecessary. Accordingly, the associated cost of services for inappropriate stay will increase as well [10]. Thus, shorter hospital LOS is usually preferred as it is associated with reduced hospital costs and expenses without affecting the patients’ outcomes [6]. This is mainly because the patient care shifts from higher inpatient costs to lower outpatient costs [9]. Apart from the financial benefits, shortening the LOS is expected to reduce other several inpatient complications, including nosocomial infections [11]. It is also expected to minimize hospital bed shortages and increase the availability of acute beds for waiting patients who must be admitted [10]. 

The primary aim of this study was to determine the hospital length of stay (LOS) of patients presenting with acute onset VTE in several public hospitals in Jordan. This study also aimed to assess the clinical profile, comorbidities, and the treatment protocols implemented, and how these factors collectively affect the LOS. 

## 2. Materials and Methods

This study is a cross-sectional, observational study that tracked acute VTE admissions between January 2019 and January 2020. This study was conducted in public hospitals in Jordan. 

### 2.1. Subjects and Data Collection

The patients included in this study were adults (age ≥ 18 years) who had been admitted to hospital and diagnosed with acute VTE. These patients were fully managed at the selected hospitals and fulfilled the diagnostic criteria for VTE, including both subjective and objective measures. 

The clinical files (case records) for the included patients were reviewed during the patients’ stay in the predetermined hospitals. Detailed data from the patients’ files and patient-reported data were also collected using a detailed survey, which was developed based on an extensive literature review. Furthermore, the survey was reviewed and corrected several times by a panel of experts with extensive clinical and research expertise. The survey included the following information: patients’ demographics (age, sex, weight, height, smoking status, job, level of education, comorbidities, family history, and city of residence), clinical features (signs and symptoms, diagnosis and diagnostic tools, and recurrence status), medical management (the anticoagulant therapies that had been offered during the hospital stay), and the duration of hospitalization and clinical outcomes (LOS and other laboratory tests). 

### 2.2. Statistical Analysis

To analyze the data, Stata^®^ statistical package version 14.0 was used [12]. The characteristics of the DVT patients were summarized in a frequency table that included information about the patients’ demographic variables, comorbidities, and medications. Hospital length of stay, the outcome of interest, was categorized into three levels: 1–3 days, 4–6 days, and ≥7 days [13]. An ordered logistic regression model was used to study the significant predictors of LOS, as the number of outcome categories exceeded two with an ordering. First, a chi-squared test, or Fisher’s exact test for variables with cells that had less than five observations, was used to study the potential significant predictors of LOS in the univariate analysis. Second, a multivariate ordered logistic model was used to generate adjusted odds ratios (AORs) and 95% confidence intervals. An alpha level of 0.05 was set as the significance level in this study. 

## 3. Results

### 3.1. Characteristics of the VTE Patients

Over a period of 12 months, a total of 317 patients admitted with VTE were included in the study out of 309 patients (8 cases are repeated admissions). Several diagnostic tools, A Doppler ultrasound and a d-dimer value, confirmed each patient’s diagnosis as a basic inclusion criterion. Table 1 summarizes the diagnostic procedures performed for patients with DVT, PE, or both. Accordingly, we were able to collect 317 cases of VTE, divided into 271 (85.8%) as DVT, 28 (8.9%) as PE, and 17 (5.4%) as both.

The characteristics of the VTE patients are summarized in Table 2. Most of the participants were male (52.4%), and 35.3% aged between 50 and 69 years. The data showed that the majority of the recorded VTE cases were admitted for the first time (64.6%). Most of the cases (57.2%) were smokers, 66.3% were overweight/obese, and 59% were hypertensive. Some less common comorbidities identified among the VTE patients included diabetes (27.4%), visible varicose vein (17.7%), vitamin D deficiency (12%), heart failure (11%), malignancy/chemotherapy (8.2%), myocardial infarction (7.6%), atrial fibrillation (7.6%), and chronic kidney disease (6.9%). Most patients were found to suffer from several (4–5) DVT symptoms, with the most common symptoms being edema (leg swellings) (*n* = 277, 87.4%) and leg pain (*n* = 264, 83.3%); tenderness (*n* = 196, 61.8%); and warmth and skin erythema (*n* = 190, 59.9%). Moreover, about 22.7% (*n*= 72) of the patients experienced shortness of breath, even without a PE diagnosis. The patients were mostly treated with one of the following: Warfarin and Enoxaparin (40.4%), Warfarin and Tinzaparin (26.5%), or Warfarin and unfractionated heparin (3.5%). The results indicated that almost no inpatients at the screened public hospitals were acutely treated with DOACs or other non-vitamin K antagonist oral anticoagulants. 

### 3.2. Predictors of Hospital Length of Stay (LOS)

Out of the total number of patients included, only 11% were found to have been admitted with VTE and to have stayed at hospital for one to three days. Patients who had stayed for four to six days formed 44% of the total number of VTE patients included, with the remaining 45% of patients having stayed for at least 7 days (Table 3). The results of the univariate analysis showed that longer hospital LOS was significantly associated with age, hypertension, diabetes mellitus, cardiovascular diseases, Warfarin and Tinzaparin treatment, visible varicose vein, and hypercoagulable state. However, according to the multivariate ordinal regression analysis (Table 4), only patients with hypertension were found to have 2.1 higher odds of a lengthened hospital LOS (95% CI: 1.18–3.7, *p* = 0.011). In contrast, patients receiving Warfarin and Tinzaparin were found to have lower odds of a lengthened hospital LOS (AOR: 0.59, 95% CI: 0.36–0.97, *p* = 0.037), as well as patients with a hypercoagulable state (AOR: 0.3, 95% CI: 0.11–0.89, *p* = 0.029)

## 4. Discussion

To our knowledge, this is the first study in Jordan to examine the characteristics of patients admitted with VTE and the predictors of hospital LOS using data from multiple hospitals. Almost half of the patients included in this study stayed for at least 7 days in the hospital. 

The most common associated comorbidities were hypertension (59%), diabetes (27%), visible varicose vein (17.7%), vitamin D deficiency (12%), and heart failure (11%). Some less common associated comorbidities included malignancy/chemotherapy (8.2%), myocardial infarction (7.6%), atrial fibrillation (7.6%), and chronic kidney disease (6.9%). Some of this study’s results are consistent with what was recorded in previous reports. A study from the USA showed that hypertension (57%), diabetes mellitus (28%), hyperlipidemia (25%), and heart failure (14%) were the most encountered comorbidities in investigated VTE patients [14]. A large, ongoing, multicenter, international, observational study called (RE-COVERY DVT/PE) enrolled 6194 consecutive VTE patients from 34 countries. The most common comorbidities detected in that clinical trial were also hypertension (34.7%), diabetes mellitus (11.4%), and cancer (excluding non-melanoma skin cancer) (11.0%). When compared to the RE-COVERY DVT/PE study results, the prevalence of hypertension and diabetes was relatively higher in our investigated sample of VTE patients [15]. Hypertension (52%) and diabetes (11.6%) were also common comorbidities observed in VTE patients managed at Spanish Emergency Departments. However, the prevalence of these comorbidities is lower than what was detected in our study [3]. Another study in an urban district hospital in KwaZulu-Natal found that HIV and TB infections were the most common comorbidities in the VTE patients of that hospital which are both uncommon infections in Jordan. It also showed heart failure (18.52%) as another common comorbidity, which is relatively higher than our findings (11%) [5]. Collectively, the prevalence of hypertension and diabetes among acutely diagnosed VTE patients in Jordan is relatively high. Accordingly, these comorbidities may play an important role in the development of VTE. To better manage VTE and reduce its recurrence, it is crucial for healthcare providers to assess the status of hypertension and diabetes control among VTE patients.

A previous meta-analysis confirmed that smoking (whether it be now or in the past) is associated with increased VTE risk [16]. The smoking rate among our investigated VTE patients (57.2%) was higher than that of patients in several other studies (25.93% [5], 15.1% [15], and 13.6% [3]). The high prevalence of smoking among VTE patients in our sample reflects the high smoking prevalence in Jordan as a whole. Recent statistics showed that smoking rates in Jordan are among the highest in the world. Healthcare providers should continue to educate VTE patients about the risks of smoking and the benefits of quitting smoking. The Jordanian government should take more serious steps towards implementing awareness campaigns and reducing the high smoking rates in Jordan. 

The rate of visible varicose veins observed in our sample (17.7%) was higher than in previous studies (8.64%, [5]). A study that tracked patients over a mean follow-up of >7 years found that the development of DVT was higher in varicose vein patients than in control patients (6.55 vs. 1.23 per 1000 person-years; hazard ratio [HR], 5.30; 95% confidence interval [CI], 5.05–5.56) [17]. Therefore, varicose vein patients should be educated about the early signs of VTE in order to prevent complications. 

Most of the VTE patients in this study were overweight/obese (66.3%). Obesity is well-recognized for being associated with an increased risk of VTE [18]. Obesity is known to increase procoagulant factors [19] and the intra-abdominal pressure, which slows down the venous return [20]. Consistent with our results, obesity was one of the most common comorbidities found by a previous study of VTE patients in Spanish Emergency Departments (31.5%) [3]. 

Regarding hospital LOS, most of this study’s hospitalized patients were treated with vitamin K antagonist (Warfarin) overlapped with low molecular weight heparins (LMWHs), including Enoxaparin and Tinzaparin. Almost no patients were offered the new non-vitamin K antagonist oral anticoagulants or DOACs as part of the routine inpatient treatment plan for acute VTE. In contrast, previous reports from other countries showed an increasing trend toward the use of new DOACs with a reduction in the use of Warfarin for VTE management. For example, a retrospective study in Brigham and Women’s Hospital showed that around 57% of patients received DOACs, whereas 41% received Warfarin [6]. Several extensive clinical trials are currently being conducted to determine the clinical advantages of using DOACs in DVT and PE patients. For example, several DOACs (Rivaroxaban, Edoxaban, Apixiban, and Dapigatran) Versus LMWH ± Warfarin for VTE in cancer were compared in NCT02744092 [21]. In the RE-COVERY DVT/PE clinical trial from 34 countries, about 54.0% (*n* = 3294) of the patients were on the new non-vitamin K antagonist oral anticoagulants (NOACs) and only 22.8% (*n* = 1388) of the patients were on vitamin K antagonist (Warfarin) [15]. The same study indicated that the use of NOACs is lower in the Middle East (21%) when compared to Europe (61%) [15]. Another clinical study investigated Rivaroxaban vs Dalteparin in cancer-associated VTE patients (NCT02746185). The efficacy and safety results were consistent with those previously reported with DOACs. However, the number of recruited patients was not sufficient [22]. The introduction of direct oral anticoagulants (DOACs) is considered one of the major advancements in VTE management in the past decade. The DOACs are an effective alternative to vitamin K antagonist (VKA) and provide an alternative oral approach without the need for parenteral medications. This approach provides the benefit of reducing the length of hospital stays and allowing earlier patient discharge [2]. A study conducted by Hisham Badreldin compared the LOS between patients receiving DOACs versus patients receiving traditional Warfarin as the treatment plan for acute VTE. The study confirmed that patients receiving DOACs had significantly shorter hospital LOS compared to those receiving Warfarin (median 3 days, [IQR 0–5] vs. 8 days [IQR 5–11], *p* < 0.05) [6]. Another study also confirmed that the hospital LOS among 78 patients who received either Apixaban or Rivaroxaban was shorter when compared to the LOS among 74 patients who received Warfarin (2.63 vs. 5.33 days; *p* < 0.05) [23]. Furthermore, VTE patients had significantly shorter LOS when treated with DOACs than when treated with Warfarin (28 vs. 114 h, *p* < 0.05), as shown in other previous studies [7,24]. A clinical trial showed that in patients with submassive PE, initiation of a DOAC (Rivaroxaban or Apixiban) shortly after catheter-directed thrombolysis may result in a decreased hospital LOS compared to parenterally bridged Warfarin (LOS: (4.0 vs 6.1 days, respectively, *p* = 0.002)) [25]. Our results showed that most patients stayed in hospital for over 3 days (89%) and about 45% of patients stayed for 7 days or more. This can be partially explained by the fact that most patients in the investigated hospitals are treated with Warfarin overlapped with injectable LMWHs. This contradicts hospitals in other countries which have recently been relying more on DOACs for a shorter treatment duration. Thus, we recommend increasing the use of DOACs instead of Warfarin. Although these medications cost more, studies have shown that the overall treatment cost is lower than when using Warfarin due to the significantly reduced hospital LOS [26,27].

The results of our univariate analysis showed that longer LOS was significantly associated with age, hypertension, diabetes mellitus, cardiovascular diseases, Warfarin and Tinzaparin treatment, visible varicose vein, and hypercoagulable state. However, according to the multivariate ordinal regression analysis, only patients with hypertension were found to have 2.1 higher odds of a lengthened hospital LOS (95% CI: 1.18–3.7, *p* = 0.011). In contrast, patients receiving Warfarin and Tinzaparin were found to have lower odds of a lengthened hospital stay, as well as patients with a hypercoagulable state. It is worth noting that the comparison in our study did not include DOACs in the analysis as no patients were using these medications in the investigated hospitals. Thus, a future study comparing the two approaches (Warfarin vs DOACs) in Jordanian patients is needed. Additionally, the multivariate ordinal regression analysis showed that it is not Warfarin alone reducing the LOS, rather it is the combination of both Warfarin and Tinzaparin compared to Warfarin with Enoxaparin or Warfarin with UFH. We do believe that if the statistical comparison included DOACs, it would result in further preferable reduction in LOS. Additionally, the variations between the two analyses are maybe due to the low sample size. Few previous studies have investigated such predictors and their correlations with hospital LOS, specifically among VTE patients. Regarding hypertension, it was also found to be associated with longer hospital LOS in a report of VTE patients from the USA [14]. Accordingly, we suggest more intensive emphasis on the prevention and control of such comorbidities. On the other hand, the predictors of hospital LOS in general were investigated extensively in previous studies. Old age and an increased number of comorbidities were found to be highly associated with increased hospital LOS [28]. A study on patients admitted for heart failure exacerbation showed that more chronic comorbidities and the severity of the disease at the time of admission are significantly associated with increased hospital LOS [13]. A prediction model for hospital LOS among cardiac patients showed that age along with systolic and diastolic pressure at the time of admission have the highest impact on the LOS [29]. According to both our current and previous results, comorbidities may have a profound impact on hospital LOS. Thus, comorbidities such as hypertension and DM should be prevented and controlled very well to avoid further complications in VTE patients and longer hospital stays. 

It is worth mentioning that the severity of DVT and PE plays an important role in determining the LOS for VTE patients. According to the latest 2019 ESC Guidelines for the diagnosis and management of acute pulmonary embolism, assessment of PE severity and risk for early death are related to several factors. Hemodynamic instability is a major factor that indicates severe PE and a high risk for early death. In PE patients with no hemodynamic instability, other factors are used, including imaging, diagnostic laboratory tools, documented right ventricular failure, as well as patients’ comorbidities [8]. Similarly, acute PE can be classified into massive or submassive PE, where massive PE can be identified if patients present with one or more of the following criteria: hypotension or shock that results from right heart failure or cardiovascular collapse, a thrombus which occludes greater than 50% of the pulmonary artery (PA) cross-sectional area or occludes two or more lobar arteries, or if the patient is dependent on inotropic agents [30]. Unfortunately, our current study could not retrieve clear data about the severity classification of DVT or PE in the investigated hospitals. Patients’ clinical records do not contain details related to the severity or the adopted classification systems used in these hospitals. Thus, public hospitals in Jordan should adopt more clear classification systems and an improved recording system for VTE patients’ assessments and follow up reports. 

This study has several limitations, including a small sample size and the presence of confounding factors that could affect the external validity; moreover, we do not have details on the use of systemic or catheter-directed thrombolysis. Additionally, as mentioned above, the severity scaling of DVT and PE could significantly affect the LOS. These factors could not be determined in the current study due to limited data available in the screened patients’ clinical records. 

Future research should focus on overcoming the current limitations of this study. A comprehensive study that includes a larger number of patients should be implemented. Furthermore, important details that could significantly affect the LOS should be included, such as the severity of thromboembolism, the degree of damage to the lung tissue, and the presence of right ventricular failure. It is well known that Warfarin is considered as one of the drugs that is causing a very frequent drug related problems [31]. Thus, future work should include a comprehensive comparison between Warfarin and DOACs for management of VTE patients. 

In conclusion, this study is a highly important descriptive study that describes the current clinical picture of acute VTE cases in several public hospitals in Jordan. Most of the VTE patients included were smokers, had a high body mass index (BMI), were hypertensive, and/or suffered from diabetes. The study demonstrated that a high ratio of hospitalized VTE patients have long hospital LOS (≥7 days). Most of the patients were treated with Warfarin and LMWHs, whilst few to no patients were offered DOACs. This can partially explain the longer hospital LOS compared to other countries. Our analysis demonstrated other predictors of long hospital stay. However, these predictors could not be used to explain the LOS in reality due to the limited number of patients and the lack of some important clinical data such as the severity of VTE for each case. The comparison in this work was not made with DOACs as no patients were using these medications in the investigated hospitals. A future study comparing the two approaches in Jordanian patients would be beneficial. We believe that if the statistical comparison included DOACs, it would result in further preferable reduction in LOS. Thus, we recommend increasing the use of DOACs instead of Warfarin in order to decrease hospital LOS. Although these medications cost more, studies have shown that the overall treatment cost is lower than when using Warfarin due to the significantly reduced hospital LOS.

## Figures and Tables

**Table 1 medicina-59-00727-t001:** The diagnostic procedures performed for patients with DVT, PE or both.

Diagnostic Procedure	Diagnosis (*n*%)		
	DVT	PE	Both
Venography	2 (0.7%)	2 (7.1%)	1 (5.9%)
MRI	3 (1.1%)	1 (3.6%)	0 (0.0%)
Ultrasound	224 (83.0%)	15 (53.6%)	10 (58.8%)
Ultrasonography	26 (9.6%)	5 (17.9%)	5 (29.4%)
Chest XR	76 (51.4%)	18 (90.0%)	10 (76.9%)
D-Dimer test	75 (27.7%)	12 (42.9%)	7 (41.2%)

DVT: Deep Vein Thrombosis, PE: Pulmonary Embolism, MRI: Magnetic Resonance Imaging.

**Table 2 medicina-59-00727-t002:** Characteristics of Patients Admitted with Venous Thromboembolism (*n* = 317).

Characteristics	*n* (%)
Age	
<40	80 (25.2%)
40–49	74 (23.3%)
50–69	112 (35.3%)
≥70	51 (16.1%)
Gender	
Male	166 (52.37%)
Female	151 (47.63%)
Diagnosis	
DVT	271 (85.8%)
PE	28 (8.9%)
Both	17 (5.4%)
Smoker	
Yes	179 (57.2%)
No	134 (42.8%)
BMI	
Underweight (<18.5)	36 (11.4%)
Healthy weight (18.5 to <25)	71 (22.4%)
Overweight/obese (≥25)	210 (66.3%)
Immobilization	
Yes	66 (21.1%)
No	247 (78.9%)
Recurrent DVT/PE	
Yes	112 (35.33%)
No	204 (64.67%)
Hypertension	
Yes	130 (59%)
No	187 (41%)
Diabetes Mellitus	
Yes	87 (27.4%)
No	230 (72.6%)
Cardiovascular Disease (MI, CHF and stroke)	
Yes	58 (18.3%)
No	259 (81.7%)
Blood transfusion (<1 month)	
Yes	21 (6.62%)
No	296 (93.4%)
Hormone Replacement Therapy or steroids	
Yes	23 (7.3%)
No	294 (92.7%)
Vit D deficiency	
Yes	38 (12%)
No	279 (88%)
Surgery/anesthesia > 45 min	
Yes	20 (6.3%)
No	297 (93.7%)
Hypothyroidism	
Yes	19 (6%)
No	298 (94%)
CKD	
Yes	22 (6.9%)
No	295 (93.1%)
Hyperlipidemia	
Yes	19 (6%)
No	298 (94%)
Hypercoagulable state	
Yes	14 (4.4%)
No	303 (95.6%)
Lung Disease	
Yes	18 (5.7%)
No	299 (94.3%)
Malignancy/Chemotherapy	
Yes	26 (8.2%)
No	291 (91.8%)
Charlson Comorbidity Index (CCI)	
CCI = 0	85 (26.8%)
CCI = 1–2	167 (52.7%)
CCI = 3–4	55 (17.4%)
CCI ≥ 5	10 (3.2%)
Warfarin + Enoxaparin	
Yes	128 (40.4%)
No	189 (59.6%)
Warfarin + Tinzaparin	
Yes	84 (26.5%)
No	191 (73.5%)
Warfarin + UFH	
Yes	11 (3.47%)
No	306 (96.5%)
Number of DVT symptoms	
0–1	39 (12.4%)
2–3	114 (36.3%)
4–5	161 (51.3%)
Presence of visible varicose vein	
Yes	56 (17.67%)
No	261 (82.3%)
Number of comorbidities	
0–1	136 (42.9%)
2–3	111 (35%)
4–5	55 (17.4%)
≥6	15 (4.73%)
Polypharmacy	
Yes	80 (25.24%)
No	237 (74.76%)

DVT: Deep Vein Thrombosis PE: Pulmonary Embolism, BMI: Body Mass Index, CHF: Congestive Heart Failure, MI: Myocardial Infarction, CKD: Chronic Kidney Disease.

**Table 3 medicina-59-00727-t003:** Significant predictors of Hospital Length of Stay (LOS) among patients with Venous Thromboembolism: Results from the univariate analysis.

Variable		LOS: 1–3 Days *n* = 34 (11%)	LOS: 4–6 Days *n* = 135 (44%)	LOS: ≥7 Days *n* = 138 (45%)	*p*-Value
Age	<40	13 (38.2%)	23 (17%)	42 (30.5%)	<0.001
	40–49	12 (35.3%)	27 (20%)	33 (23.9%)	
	50–69	9(26.5%)	54 (40%)	45 (32.6%)	
	≥70	0	31 (23%)	18 (13%)	
Hypertension	Yes	7 (20.6%)	55 (49.7%)	64 (46.4%)	0.019
	No	27 (79.4%)	80 (59.3%)	74 (53.6%)	
Diabetes Mellitus	Yes	3 (8.8%)	42 (31.1%)	39 (28.3%)	0.023
	No	31 (91.2%)	93 (68.9%)	99 (71.7%)	
Cardiovascular Diseases	Yes	3 (8.8%)	34 (25.2%)	20 (14.5%)	0.028
	No	31 (91.2%)	101 (74.8%)	118 (85.5%)	
Warfarin + Tinzaparin	Yes	8 (23.5%)	46 (34.1%)	28 (20.3%)	0.033
	No	26 (76.5%)	89 (65.9%)	110 (79.7%)	
Visible Varicose Vein	Yes	4 (11.8%)	34 (25.2%)	17 (12.3%)	0.014
	No	30 (88.2%)	101 (74.8%)	121 (87.7%)	
Hypercoagulable state	Yes	3 (8.8%)	8 (5.9%)	2 (1.5%)	0.03
	No	31 (91.2%)	127 (94.1%)	136 (98.5%)	

Fisher’s exact test was used for variables with cells that have *n* < 5.

**Table 4 medicina-59-00727-t004:** Predictors of Hospital Length of Stay (LOS) among Patients with Venous Thromboembolism Using Multivariate Ordinal Logistic Regression.

Variables		Adjusted Odds Ratio (AOR)	Standard Error (SE)	*p*-Value	95% Confidence Interval
Age	<40	Ref			
	40–49	0.64	0.22	0.179	0.32–1.23
	50–69	0.57	0.18	0.075	0.30–1.1
	≥70	0.48	0.19	0.067	0.22–1.1
Hypertension	No	Ref			
	Yes	2.1	0.60	0.011	1.18–3.70
Diabetes Mellitus	No	Ref			
	Yes	1.07	0.31	0.81	0.6–1.9
Cardiovascular Diseases	No	Ref			
	Yes	0.69	0.20	0.207	0.38–1.23
Warfarin + Tinzaparin	No	Ref			
	Yes	0.59	0.15	0.037	0.36–0.97
Visible Varicose Vein	No	Ref			
	Yes	0.82	0.23	0.498	0.47–1.44
Hypercoagulable state	No	Ref			
	Yes	0.30	0.17	0.029	0.11–0.89

## Data Availability

Available upon request from corresponding author.

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
