# Peer review of "The Length of Hospital Stay of Patients with Venous Thromboembolism: A Cross-Sectional Study from Jordan"

_medicina, 2023, doi:10.3390/medicina59040727_

Round 1
Reviewer 1 Report
The manuscript submitted for review by Amawi et al., "The Length of Hospital Stay of Patients with Venous Thromboembolism: A Cross-sectional Study from Jordan", analyzes the factors that determine Length of Hospital Stay of Patients with Venous Thromboembolism in Jordanian hospitals. The authors also tried to present a "portrait" of a patient with Venous Thromboembolism in hospitals in Jordan. This is probably important for the medical institutions of this region, however, while reading the manuscript, I had a number of comments and questions that I would like to receive answers from the authors.
1. First of all, the main indicator studied by the authors, Length of Hospital Stay, raises questions. It is clear that, first of all, the duration of treatment depends primarily on the severity of thromboembolism, the degree of damage to the lung tissue and the presence of right ventricular failure. The authors of the article did not study any of these indicators, so it seems that they studied random factors that did not determine the duration of hospitalization per se. In fact, for patients with venous thromboembolism, other factors that determine the results of treatment are much more important - deaths, the frequency of repeated thromboembolism, the frequency of bleeding.
2. The indicators that the authors used to characterize their patients also raise questions. In my opinion, the authors did not take into account a number of important indicators - the presence of a tumor, what was the initial risk of venous thrombosis and thromboembolism during operations, prolonged immobilization, strokes, whether thromboprophylaxis was performed, etc. Without these data, patient information is clearly incomplete. In addition, there are no data on diagnostic methods for confirming the diagnosis (angiopulmonography data, lung scanning, d-dimer level, etc.).
3. As a consequence, the conclusions reached by the authors look strange. For example, they state that "Our analysis demonstrated that other predictors of long hospital stay include hypertension, diabetes mellitus, cardiovascular diseases, warfarin and tinzaparin treatment, visible varicose vein, and hypercoagulable state". In fact (judging by the data in Table 3), of these factors, only arterial hypertension is associated with LOC (RR 2.1), while other indicators either do not have an independent association with LOC (diabetes mellitus, Cardiovascular Diseases, Visible Varicose Vein), or, conversely, are associated with a shorter duration LOC (for Warfarin + Tinzaparin OR is 0.59, for Hypercoagulable state - 0.30). These data are difficult to explain and understand, perhaps due to errors in statistical calculations.
4. In general, statistical analysis is poorly described, it is not clear what was the dependent binary variable in multiple logistic regression analysis, the authors did not indicate this.
5. Table 1 indicates that Deep Vein Thrombosis was diagnosed in 85.8% of patients, Pulmonary Embolism in 8.9%, and Both in 5.3% of patients. How does this compare to the statement in the Subjects and Data Collection section that "The patients included in this study were adults (age ≥18 years) who had been admitted to hospital and diagnosed with acute VTE"?
6. Did the authors leave the paragraph in the Discussion section (lines 258-261) from the recommendations to authors?
Author Response
Dear respected reviewer,
We are really thankful for your helpful comments. we really understand your concerns so we tried our best to improve the presentation of the results to meet your standards. However, please note that this observational study provided important update about the current clinical picture for the management of acute VTE cases in Jordan. It shows the most likely comorbidities that are associated with VTE and the current lack of clear severity classification system for VTE cases. the current study also shows the lower use of DOACs compared to other areas in the world. I hope you like the updated version of the manuscript. Also, Please note that we added a new table to show the most common diagnostic tools adopted in the investigated hospitals.

Reviewer 2 Report
I read with great interest the paper entitled “The Length of Hospital Stay of Patients with Venous Thrombo-embolism: A Cross-sectional Study from Jordan” authored by Amawi et al.
I really trust that the direct oral anticoagulant (DOAC) therapy is a keystone in the management of pulmonary embolism and, also due to the good safety and efficacy profile, it is plausible that the use of DOACs could be facilitate the clinical management of the patients.
However, in my humble opinion, there are some methodological and material errors in the paper and, in order to make this work passable for publication I would like to suggest some revisions:
1. Please add the background in the abstract
2. In the introduction at line 61, please cite and comment the latest European Society of Cardiology guidelines about Pulmonary Embolism published in 2019 (European Heart Journal, Volume 41, Issue 4, 21 January 2020, Pages 543–603)
3. At line 75 please change “venous thromboembolism” with “VTE” and verify the possibility of use the same acronym in all the text, where indicated.
4. In Materials and Methods, please specify how the diagnosis of pulmonary embolism was made.
5. As correctly described in the introduction venous thromboembolism (VTE) is a cardiovascular disease that can manifest as deep vein thrombosis (DVT), pulmonary embolism (PE) or both. At line 111 in the Results there is “Characteristics of the DVT Patients” and 317 patients with DVT are described. Consistent with what was announced in the Abstract and in the Introduction, the 317 patients should have VTE in terms of DVP with or without PE. These affirmations are in contrast.
Please describe in the Results how many patients have a VTE and specify how many patients have DVP, PE or both.
6. This mistake influences the scientific soundness of all paper because the length of hospital stay is heavily influenced by the presence of PE.
7. Moreover, in the management of pulmonary embolism it is necessary to classify the disease in low, medium and high risk of death (as reported in ESG guideline 2019). E.g.: A patient with low risk PE is easily discharged in a few days compared with patients with high risk.
8. Finally, I would like to remind the authors to describe the main clinical trials involving the treatment of pulmonary embolism with DOACs and to indicate the posology (dose loading etc) and the contraindications to this type of therapy (renal failure, liver failure, drug-drug interactions etc)
In conclusion, I really think that the manuscript treats a very interesting topic, but needs major methodological revision before the publication.
I would like to incite the authors to improve what has already been done as it will surely come out a great work!
Author Response

(The authors gave the same response as above.)

Round 2
Reviewer 1 Report
The authors have significantly improved the article, but I am not entirely satisfied with the answer to my question 3.
In Table 3, according to the Multivariate Ordinal Logistic Regression data, two factors reduce LOS - Warfarin + Tinzaparin (OR is 0.59), and Hypercoagulable state (OR 0.30). In conclusion, the authors suggest using "DOACs instead of warfarin in order to decrease hospital LOS". I don't see the logic in this proposal - judging by the authors' data, the use of warfarin already reduces the duration of hospitalization. Should it be further reduced? Also, the state of hypercoagulability according to the authors reduces LOS. So maybe you don’t have to fight him, since his action is so favorable? The manuscript contains no reasoning of the authors regarding the data of their results. One gets the impression that first of all they want to justify the use of DOAC in patients with VTE, which they are trying to do, despite their own results.
Author Response
Dear Respected reviewer,
We are really thankful for your efforts in reviewing and improving this manuscript:
Reviewer #1:
The authors have significantly improved the article, but I am not entirely satisfied with the answer to my question 3.
Response: We thank the reviewer for his comments. We are really thankful for your helpful comments. We are really glad that you think the manuscript is highly improved now. We will try to improve more as needed.
In Table 3, according to the Multivariate Ordinal Logistic Regression data, two factors reduce LOS - Warfarin + Tinzaparin (OR is 0.59), and Hypercoagulable state (OR 0.30). In conclusion, the authors suggest using "DOACs instead of warfarin in order to decrease hospital LOS". I don't see the logic in this proposal - judging by the authors' data, the use of warfarin already reduces the duration of hospitalization. Should it be further reduced?
Response: We thank the reviewer for his comments. The conclusion was modified in more proper language. Please note that the comparison here was not made with DOACs as no patients are using these medications in the investigated hospitals. A future study comparing the two approaches in Jordanian patients would be really great. Please also note that it is not warfarin alone that found to reduce the LOS, it is the combination of both warfarin and Tinzaparin compared to for example warfarin with enoxaparin or warfarin with UFH. And yes we do believe that if the statistical comparison included DOACs, it would result in further preferable reduction in hospital stay. This will be included in the discussions and conclusion.
Also, the state of hypercoagulability according to the authors reduces LOS. So maybe you don’t have to fight him, since his action is so favorable? The manuscript contains no reasoning of the authors regarding the data of their results. One gets the impression that first of all they want to justify the use of DOAC in patients with VTE, which they are trying to do, despite their own results
Response: We thank the reviewer for his comments. The presence of hypercoagulability for sure increase the risk of VTE incidence and there is a huge body of data confirming this fact. However, it is not necessarily should increase the length of hospital stay after the development of the VTE event. In fact this may sound right as many patients of hypercoagulability status are otherwise healthy and may have no other comorbidities which may explain the shorter hospital stay. Again, this doesn’t mean that it is a good factor that we should not fight. It just mean that it may not result in longer hospital stay.

Reviewer 2 Report
I have greatly appreciated the improvements made to this work, which, in my opinion, can now be considered for publication.
I only report below some formal corrections that should be made:
1. The subtitle 3.1 at line 122 should be changed in: "3.1. Characteristics of the VTE Patients"
2. Table 1 and table 2 should be edited as request by author guidelines.
3. The bibliography should be enumerate and cite in the text as required by the author guidelines
Overall, the work is very interesting and I hope it will also be judged positively by the editorial board.
Author Response
1-I have greatly appreciated the improvements made to this work, which, in my opinion, can now be considered for publication.
Response: We thank the reviewer for the comments. We are really glad that you appreciated the improvements made to this work and you find it acceptable for publication.
Only report below some formal corrections that should be made:
- The subtitle 3.1 at line 122 should be changed in: "3.1. Characteristics of the VTEPatients"
Response: We thank the reviewer for the comment. The subtitle was changed as suggested.
- Table 1 and table 2 should be edited as request by author guidelines.
Response: We thank the reviewer for the comment. The tables formatting was changed according to authors guidelines.
- The bibliography should be enumerate and cite in the text as required by the author guidelines
Response: We thank the reviewer for the comment. The references were corrected according to the author guidelines.
- Overall, the work is very interesting and I hope it will also be judged positively by the editorial board.
Response: Thank you very much and we are really thankful for all your help and efforts to improve this manuscript.
